# Reinforcement Learning with Multiple Experts: A Bayesian Model Combination Approach

**Michael Gimelfarb**
Mechanical and Industrial Engineering
University of Toronto
mike.gimelfarb@mail.utoronto.ca

**Scott Sanner**
Mechanical and Industrial Engineering
University of Toronto
ssanner@mie.utoronto.ca

**Chi-Guhn Lee**
Mechanical and Industrial Engineering
University of Toronto
cglee@mie.utoronto.ca

## Abstract

Potential based reward shaping is a powerful technique for accelerating convergence of reinforcement learning algorithms. Typically, such information includes an estimate of the optimal value function and is often provided by a human expert or other sources of domain knowledge. However, this information is often biased or inaccurate and can mislead many reinforcement learning algorithms. In this paper, we apply Bayesian Model Combination with multiple experts in a way that learns to trust a good combination of experts as training progresses. This approach is both computationally efficient and general, and is shown numerically to improve convergence across discrete and continuous domains and different reinforcement learning algorithms.

## 1 Introduction

*Potential-based reward shaping* incorporates prior domain knowledge in the form of additional rewards provided during training to speed up convergence of reinforcement learning algorithms, without changing the optimal policies (Ng et al. [1999]). While much of the existing theory and applications assume that advice comes from a single source throughout training (Grześ [2017], Harutyunyan et al. [2015], Tenorio-Gonzalez et al. [2010]), there is much less work done on learning from multiple sources of advice as training progresses. One reason for doing so is that expert demonstrations or advice can often be biased or incomplete, so being able to identify good advice from bad is critical to guarantee robustness of convergence.

In this paper, the decision maker is presented with multiple sources of expert advice in the form of potential-based reward functions, some of which can be misleading and should not be trusted. The decision maker does not know a priori which expert(s) to trust, but rather learns this from experience in a Bayesian framework. More specifically, the decision maker starts with a *prior* distribution over the probability simplex, and updates the belief to a *posterior* distribution as new training rewards are observed. Because our proposed algorithm follows the potential-based reward shaping framework, it preserves the theoretical guarantees for policy invariance established in Ng et al. [1999].

This paper proceeds as follows. Section 2 introduces the key definitions used throughout the paper. In Section 3, we apply *Bayesian model combination*, that allows the decision maker to asymptotically learn the best combination of experts, all with reduced variance as compared to similar approaches. In Section 3.1, we show that the total return can be written as a linear combination of individual return

contributions from each expert, weighted by the expected posterior belief that the expert is correct. In Section 3.2, we show that the exact posterior updates are analytically intractable. Instead, we apply *moment matching* to project the true posterior distribution onto the multivariate Dirichlet distribution, and show how accurate approximation and inference can be done in linear time in the number of experts. In Section 3.3 we then show how our approach can be incorporated into any reinforcement learning algorithm, preserving the asymptotically optimal policy without incurring additional runtime complexity. Finally, in Section 4, we demonstrate the effectiveness of this approach across various reinforcement learning methods and problem domains.

**Related Work**

Learning from expert knowledge is not new. In *transfer learning*, for example, the decision maker uses prior knowledge obtained from training on task(s) to improve performance on future tasks (Konidaris and Barto [2006]). In *inverse reinforcement learning*, the agent recovers an unknown reward function that can then be used for shaping (Suay et al. [2016]). In many cases, a human expert can directly provide the learning agent with training examples or preferences before or during training to guide exploration (Brys et al. [2015], Christiano et al. [2017]). All of these approaches try to perturb the intermediate value functions to encourage more guided exploration of the state space. A somewhat different approach, called *policy shaping*, instead reshapes the learned policies (Griffith et al. [2013]). Grzes and Kudenko [2009] and Grześ and Kudenko [2010] recently introduced on-line methods to learn a reward shaping function, but only for model-based learning using R-Max or model-free learning with multi-grid discretization. Our approach can work in on-line settings, with general algorithms under minimal assumptions, and with value function approximation.

The idea of combining multiple models/experts or learning algorithms to improve performance is central to *ensemble learning* (Dietterich [2000]), and has been applied in a variety of ways in the RL literature. For example, Maclin et al. [2005] used kernel regression, Philipp and Rettinger [2017] used contextual bandits, and Downey and Sanner [2010] applied Bayesian model averaging. Asmuth et al. [2009] applied a Bayesian method to sample multiple models for action selection. The only work we are aware of that incorporated reward shaping advice in a Bayesian learning framework is the recent paper by Marom and Rosman [2018]. However, that paper exploited the structure of the transition model (belief clusters) in order to do efficient Bayesian inference, whereas our paper focuses on posterior approximation using variational ideas, and their analysis and results are considerably different from ours. More generally, Bayesian approaches have many advantages over frequentist approaches, including prior specification, clear and intuitive interpretation, ability to test hypotheses (O'Hagan [2004]), and theoretically optimal exploration (Thompson [1933]).

## 2 Definitions

### 2.1 Markov Decision Process

The decision-making framework used throughout this paper focuses on the *Markov decision process (MDP)* (Bertsekas and Shreve [2004]). Formally, an MDP is defined as a tuple $(\mathcal{S}, \mathcal{A}, T, R, \gamma)$, where: $\mathcal{S}$ is a general set of states, $\mathcal{A}$ is a finite set of actions, $T : \mathcal{S} \times \mathcal{A} \times \mathcal{S} \to \mathbb{R}_+$ is a stationary Markovian transition function, where $T(s, a, s') = \mathbb{P}(s'|s, a)$, the probability of transitioning to state $s'$ after taking action $a$ in state $s$, $R : \mathcal{S} \times \mathcal{A} \times \mathcal{S} \to \mathbb{R}$ is a bounded reward function, where $R(s, a, s')$ is the immediate reward received upon transitioning to state $s'$ after taking action $a$ in state $s$, and $\gamma \in [0, 1]$ is a discount factor.

We define a random policy $\mu$ as a probability distribution $\mu(s, a) = \mathbb{P}(a|s)$ over actions $\mathcal{A}$ given current state $s$. Given an MDP $(\mathcal{S}, \mathcal{A}, T, R, \gamma)$, a policy $\mu$, and initial state-action pair $(s, a)$, we define the infinite-horizon expected discounted rewards as

$$Q^\mu(s, a) = \mathbb{E}\left[\sum_{t=0}^{\infty} \gamma^t R(s_t, a_t, s_{t+1}) \middle| s_0 = s, a_0 = a\right], \tag{1}$$

where $a_t \sim \mathbb{P}(\cdot|s_t) = \mu(s_t, \cdot)$ and $s_{t+1} \sim T(s_t, a_t, \cdot)$. The objective of the agent is to find an optimal policy $\mu^*$ that maximizes (1). When the transition and reward functions are known, the existence of an optimal deterministic stationary policy is guaranteed, in which case *value iteration* or *policy iteration* can be used to find an optimal policy (Bertsekas and Shreve [2004]).

## 2.2 Reinforcement Learning

In the *reinforcement learning (RL)* setting, the transition probabilities and reward function are not explicitly known to the agent but rather learned from experience. In order to learn the optimal policies in this framework and to facilitate the development of the paper, we follow *generalized policy iteration* (GPI) (Sutton and Barto [2018]). However, the Bayesian framework developed in this paper is dependent on neither the exploration policy used nor the value function representation, and can be applied with on-policy and off-policy learning, value function approximation, traces, deep RL (Li [2017]), and other approaches.

More specifically, GPI performs two steps in alternation: a *policy evaluation* step that estimates the value $Q^{\mu_i}$ of the current policy $\mu_i$, and a *policy improvement* step that uses $Q^{\mu_i}$ to construct a new policy $\mu_{i+1}$. In practice, these two steps are often interleaved. A simple yet effective way to implement GPI is to follow the *$\varepsilon$-greedy* policy, that encourages exploration by randomly and uniformly selecting an action in $\mathcal{A}$ at time $t$ with probability $\varepsilon_t$, and otherwise selects the best action based on $Q^{\mu_i}(s, a)$; the parameter $\varepsilon_t \in (0, 1), t \geq 0$ controls the trade-off between exploration and exploitation.

In order to estimate the value of policy $\mu = \mu_i$, we follow the *temporal difference learning* (TD) approach. Specifically, given a new estimate of the expected future returns $R_t$ at time $t$ after taking action $a_t$ in state $s_t$ according to some policy $\mu$, $Q(s_t, a_t)$ (dropping the dependence on $\mu$) is updated as follows

$$Q_{t+1}(s_t, a_t) = Q_t(s_t, a_t) + \alpha \left[ R_t - Q_t(s_t, a_t) \right], \qquad (2)$$

where $\alpha > 0$ is a problem-dependent learning rate parameter.

Two popular approaches for estimating $R_t$ are *Q-learning* and *SARSA*, given respectively as

$$R_t = r_t + \gamma \max_{a' \in \mathcal{A}} Q_t(s_{t+1}, a')$$
$$R_t = r_t + \gamma Q_t(s_{t+1}, a_{t+1}), \qquad (3)$$

where $r_t = R(s_t, a_t, s_{t+1})$ is the immediate reward, $s_{t+1} \sim T(s_t, a_t, \cdot)$ and $a_{t+1} \sim \mu(s_{t+1}, \cdot)$. While both approaches compute $R_t$ by bootstrapping from current Q-values, the key distinction between them is that SARSA is an *on-policy* algorithm whereas Q-learning is *off-policy*. $n$-step TD and TD($\lambda$) are more sophisticated examples of TD-learning algorithms (Sutton and Barto [2018]).

## 2.3 Potential-Based Reward Shaping

In many domains, particularly when rewards are sparse and the agent cannot learn quickly, it is necessary to incorporate prior knowledge in order for TD-learning to converge faster. The idea of *reward shaping* is to incorporate prior knowledge about the domain in the form of additional rewards during training to speed up convergence towards the optimal policy. Formally, given an MDP $(\mathcal{S}, \mathcal{A}, T, R, \gamma)$ and a *reward shaping function* $F : \mathcal{S} \times \mathcal{A} \times \mathcal{S} \to \mathbb{R}$, we solve the MDP $(\mathcal{S}, \mathcal{A}, T, R', \gamma)$ with reward function $R'$ given by

$$R'(s, a, s') = R(s, a, s') + F(s, a, s'). \qquad (4)$$

While this approach has been applied successfully to many problems, an improper choice of shaping function can change the optimal policy (Randløv and Alstrøm [1998]).

In order to address this problem, *potential-based reward shaping* was proposed, in which $F$ is restricted to functions of the form

$$F(s, a, s') = \gamma \Phi(s') - \Phi(s), \qquad (5)$$

where $\Phi : \mathcal{S} \to \mathbb{R}$ is called the *potential function*. It has been shown that this is the only class of reward shaping functions that preserves policy optimality (Ng et al. [1999]). Reward shaping has also been shown to be equivalent to Q-value initialization (Wiewiora [2003]). More recently, policy invariance has been extended for non-stationary time-dependent potential functions of the form

$$F(s, a, t, s', t') = \gamma \Phi(s', t') - \Phi(s, t) \qquad (6)$$

(Devlin and Kudenko [2012]), for action-dependence (Wiewiora et al. [2003]), as well as for partially-observed (Eck et al. [2013]) and multi-agent systems (Devlin and Kudenko [2011]).

# 3 Bayesian Reward Shaping

The decision maker is presented with advice from $N \geq 1$ experts in the form of potential functions $\Phi_1, \Phi_2, \ldots \Phi_N$. The advice could come from heuristics or guesses (Harutyunyan et al. [2015]), from similar solved tasks (Taylor and Stone [2009]), from demonstrations (Brys et al. [2015]), and in general can be analytic or computational. One concrete example that our proposed setup can be applied to is *transfer learning* (Taylor and Stone [2009]). Here, models are first trained on a number of tasks to obtain corresponding value functions. By defining suitable *inter-task* mappings (Taylor et al. [2007]), these value functions can be incorporated into a target task as reward shaping advice.

Unfortunately in practice, the advice available to the learning agent is often contradictory or contains numerical errors, in which case it could hurt convergence. In order to make optimal use of the expert advice during the learning process, the agent should ideally learn which expert(s) to trust as more information becomes available, and act on this knowledge by applying the techniques in Section 2.2. To do this, the agent assigns weights $\mathbf{w}$ to the experts and updates them on-line during training.

The two main approaches to incorporating multiple models in a Bayesian framework are *Bayesian model averaging (BMA)* and *Bayesian model combination (BMC)*. Roughly speaking, taking experts as hypotheses, BMA converges asymptotically toward the optimal *hypothesis*, while BMC converges toward the optimal *ensemble*. The model combination approach has two clear advantages over model averaging: (1) when two or more potential functions are optimal, it will converge to a linear combination of them, and (2) it provides an estimator with reduced variance (Minka [2000]). In this section, we show how BMC can be used to incorporate imperfect advice from multiple experts into reinforcement learning problems, all with the same space and time complexity as TD-learning.

## 3.1 Bayesian Model Combination

In the general setting of Bayesian model combination, we interpret Q-values for each state-action pair $q_{s,a}$ as random variables, and maintain a set of past return observations $\mathcal{D}$ and a multivariate posterior probability distribution $P(\mathbf{q}|\mathcal{D})$ over Q-values. We also maintain a posterior probability distribution $\pi : \mathcal{S}^{N-1} \rightarrow \mathbb{R}_+$ over the $(N-1)$-dimensional probability simplex $\mathcal{S}^{N-1} = \left\{ \mathbf{w} \in \mathbb{R}^N : \sum_{i=1}^{N} w_i = 1, \, w_i \geq 0 \right\}$. Here, weight vectors $\mathbf{w}$ are interpreted as categorical distributions over experts; such a mechanism will allow us to learn the optimal *distribution* over experts, rather than a single expert. In the following subsections, we show how to maintain each of these distributions over time, but here we show how to use them for the general RL problem.

Given a state $s = s_t$ and action $a = a_t$ at time $t$, the return under model combination $\rho_t(s, a)$ is

$$\rho_t(s, a) = \mathbb{E}\left[q_{s,a}|\mathcal{D}\right] = \int_{\mathbb{R}} q \, \mathbb{P}\left(q|\mathcal{D}\right) \mathrm{d}q$$

$$= \int_{\mathbb{R}} q \int_{\mathcal{S}^{N-1}} \mathbb{P}\left(q|\mathcal{D}, \mathbf{w}\right) \mathbb{P}\left(\mathbf{w}|\mathcal{D}\right) \mathrm{d}\mathbf{w} \, \mathrm{d}q = \int_{\mathbb{R}} q \int_{\mathcal{S}^{N-1}} \sum_{i=1}^{N} \mathbb{P}\left(q|i\right) w_i \pi_t(\mathbf{w}) \, \mathrm{d}\mathbf{w} \, \mathrm{d}q \quad (7)$$

$$= \sum_{i=1}^{N} \int_{\mathbb{R}} q \, \mathbb{P}\left(q|i\right) \int_{\mathcal{S}^{N-1}} w_i \pi_t(\mathbf{w}) \, \mathrm{d}\mathbf{w} \, \mathrm{d}q = \sum_{i=1}^{N} \int_{\mathbb{R}} q \, \mathbb{P}\left(q|i\right) \mathbb{E}_{\pi_t}\left[w_i\right] \mathrm{d}q \quad (8)$$

$$= \sum_{i=1}^{N} \mathbb{E}_{\pi_t}\left[w_i\right] \int_{\mathbb{R}} q \, \mathbb{P}\left(q|i\right) \mathrm{d}q = \sum_{i=1}^{N} \mathbb{E}_{\pi_t}\left[w_i\right] \mathbb{E}\left[q_{s,a}|i\right], \quad (9)$$

where: the first equality in (7) follows from the law of total probability applied to $\mathbb{P}\left(q|\mathcal{D}\right)$, whereas the second equality follows from conditioning on the expert $i \in \{1, 2 \ldots N\}$, using the facts that $q_{s,a}$ is independent of $\mathbf{w}$ given $i$ and $\mathbb{P}\left(i|\mathbf{w}\right) = w_i$; the first equality in (8) follows from interchange of summation and integration, while the second from the definition of expectation over $w_i$; finally, (9) follows from the definition of expectation of $q_{s,a}$ given $i$.

This result is intuitively and computationally pleasing, and shows that *the total return can be written as a linear combination of individual return "contributions" from each expert model, weighted by the expected posterior belief that the expert is correct*. We now show how each of these two expectations can be computed.

## 3.2 Posterior Approximation using Moment Matching

Starting with prior distribution $\pi_t$ at time $t$ over the simplex $\mathcal{S}^{N-1}$, and given new data point $d$, we would like to perform a posterior update using *Bayes' theorem*

$$\pi_{t+1}(\mathbf{w}) = \mathbb{P}\left(\mathbf{w}|\mathcal{D}, d\right) \propto \mathbb{P}\left(d|\mathbf{w}\right)\pi_t(\mathbf{w}) \propto \sum_{i=1}^{N} \mathbb{P}\left(d|i\right) \mathbb{P}\left(i|\mathbf{w}\right)\pi_t(\mathbf{w})$$

$$= \frac{1}{C_{t+1}} \sum_{i=1}^{N} e_i w_i \pi_t(\mathbf{w}), \tag{10}$$

where we denote evidence $e_i = \mathbb{P}\left(d|i\right)$, and $C_{t+1}$ is the normalizing constant for $\pi_{t+1}$ determined as

$$C_{t+1} = \int_{\mathcal{S}^{N-1}} \sum_{i=1}^{N} e_i w_i \pi_t(\mathbf{w})\,\mathrm{d}\mathbf{w} = \sum_{i=1}^{N} e_i \int_{\mathcal{S}^{N-1}} w_i \pi_t(\mathbf{w})\,\mathrm{d}\mathbf{w} = \sum_{i=1}^{N} e_i\, \mathbb{E}_{\pi_t}\left[w_i\right]. \tag{11}$$

Unfortunately the exact posterior update is computationally intractable for general evidence $e_i$, and so an approximate posterior update is required.

*Assumed density filtering*, or *moment matching*, projects the true posterior distribution $\pi_{t+1}$ onto an exponential subfamily of proposal distributions by minimizing the KL-divergence between $\pi_{t+1}$ and the proposal distribution. We note that an excellent exponential family proposal distribution for our posterior in (10) is the multivariate *Dirichlet distribution* with parameters $\boldsymbol{\alpha} \in \mathbb{R}_+^N$, density function

$$f(\mathbf{w}; \boldsymbol{\alpha}) = \frac{\Gamma\left(\sum_{i=1}^{N} \alpha_i\right)}{\prod_{i=1}^{N} \Gamma(\alpha_i)} \prod_{i=1}^{N} w_i^{\alpha_i - 1}, \ \mathbf{w} \in \mathcal{S}^{N-1}, \tag{12}$$

and generalized moments

$$\mathbb{E}_f\left[\prod_{i=1}^{N} w_i^{n_i}\right] = \frac{\Gamma\left(\sum_{i=1}^{N} \alpha_i\right)}{\Gamma\left(\sum_{i=1}^{N}(\alpha_i + n_i)\right)} \prod_{i=1}^{N} \frac{\Gamma(\alpha_i + n_i)}{\Gamma(\alpha_i)}, \ n_i \geq 0. \tag{13}$$

For the exponential family of proposal distributions, exact moment matching requires the moments over the sufficient statistics. Since this is not available for the Dirichlet family in closed form, it necessitates an iterative approach that is not computationally feasible in on-line RL. Instead, we follow Hsu and Poupart [2016] and Omar [2016] by matching the moments (13), leading to an efficient closed-form $O(N)$ time update.

In particular, given means $m_1, m_2 \ldots m_{N-1}$ of marginals $w_1, w_2 \ldots w_{N-1}$ of $\pi_{t+1}$ and second moment $s_1$ of $w_1$, we apply approximate moment matching with proposal $\text{Dir}(\boldsymbol{\alpha})$ by solving the system of equations

$$m_i = \frac{\alpha_i}{\alpha_0}, \ i = 1, 2 \ldots N - 1 \tag{14}$$

$$s_1 = \frac{\alpha_1(\alpha_1 + 1)}{\alpha_0(\alpha_0 + 1)} \tag{15}$$

where $\alpha_0 = \sum_{i=1}^{N} \alpha_i > 0$. Please note that the second moment condition (15) is necessary here, since without it the system is under-determined. Also, we could use any of $s_2, s_3, \ldots s_N$ in place of $s_1$; in our experiments, we use the value of $s_i$ which results in the largest value of $s_i - m_i^2$ to avoid underflow in the solution. The unique positive solution of (14) and (15) is

$$\alpha_0 = \frac{m_1 - s_1}{s_1 - m_1^2}$$
$$\alpha_i = m_i \alpha_0 = m_i \left(\frac{m_1 - s_1}{s_1 - m_1^2}\right), \ i = 1, 2 \ldots N - 1. \tag{16}$$

In order to apply the moment matching solution (16) to approximate the posterior update (10), it remains to compute the moments $m_1, m_2, \ldots m_{N-1}$ and $s_1$ of $\pi_{t+1}$.

We proceed by induction on $t$. More specifically, we assume that the prior $\pi_0 = \mathrm{Dir}(\boldsymbol{\alpha}_0)$ was chosen arbitrarily and that the projection $\mathrm{Dir}(\boldsymbol{\alpha}_t)$ of $\pi_t$ was already obtained. Given new evidence $\mathbf{e} \in \mathbb{R}_+^N$, we obtain $C_{t+1} = \sum_{i=1}^N e_i \, \mathbb{E}_{\pi_t}[w_i] = \sum_{i=1}^N e_i \frac{\alpha_{t,i}}{\alpha_{t,0}} = \frac{\mathbf{e} \cdot \boldsymbol{\alpha}_t}{\alpha_{t,0}}$ where $\alpha_{t,0} = \sum_{i=1}^N \alpha_{t,i} > 0$. Using (10) and (13),

$$
\begin{aligned}
m_i = \mathbb{E}_{\pi_{t+1}}[w_i] &= \int_{\mathcal{S}^{N-1}} \frac{\alpha_{t,0}}{\mathbf{e} \cdot \boldsymbol{\alpha}_t} \sum_{j=1}^N e_j w_j w_i \pi_t(\mathbf{w}) \, \mathrm{d}\mathbf{w} \\
&= \frac{\alpha_{t,0}}{\mathbf{e} \cdot \boldsymbol{\alpha}_t} \sum_{j=1}^N e_j \int_{\mathcal{S}^{N-1}} w_j w_i \pi_t(\mathbf{w}) \, \mathrm{d}\mathbf{w} = \frac{\alpha_{t,0}}{\mathbf{e} \cdot \boldsymbol{\alpha}_t} \sum_{j=1}^N e_j \, \mathbb{E}_{\pi_t}[w_i w_j] \\
&= \frac{\alpha_{t,0}}{\mathbf{e} \cdot \boldsymbol{\alpha}_t} \left( e_i \, \mathbb{E}_{\pi_t}[w_i^2] + \sum_{j \neq i} e_j \, \mathbb{E}_{\pi_t}[w_i w_j] \right) \\
&= \frac{\alpha_{t,0}}{\mathbf{e} \cdot \boldsymbol{\alpha}_t} \left( e_i \frac{\alpha_{t,i}(\alpha_{t,i}+1)}{\alpha_{t,0}(\alpha_{t,0}+1)} + \sum_{j \neq i} e_j \frac{\alpha_{t,i}\alpha_{t,j}}{\alpha_{t,0}(\alpha_{t,0}+1)} \right) \\
&= \frac{\alpha_{t,i}(e_i + \mathbf{e} \cdot \boldsymbol{\alpha}_t)}{(\mathbf{e} \cdot \boldsymbol{\alpha}_t)(\alpha_{t,0}+1)}.
\end{aligned}
\tag{17}
$$

Using the same technique, we can readily obtain the corresponding formula for $s_1$,

$$
s_1 = \frac{\alpha_{t,1}(\alpha_{t,1}+1)(2e_1 + \mathbf{e} \cdot \boldsymbol{\alpha}_t)}{(\mathbf{e} \cdot \boldsymbol{\alpha}_t)(\alpha_{t,0}+1)(\alpha_{t,0}+2)}.
\tag{18}
$$

Combining (17) and (18) with the general solution to the moment matching problem (16) yields the new projected posterior $\mathrm{Dir}(\boldsymbol{\alpha}_{t+1})$. This leads to a very efficient $O(N)$ algorithm for posterior updates given in Algorithm 1.

---

**Algorithm 1** PosteriorUpdate($\boldsymbol{\alpha}_t, \mathbf{e}$)

---

1: **for** $i = 1, 2 \ldots N-1$ **do**   $\triangleright$ Compute posterior moments
2:   $m_i \leftarrow \frac{\alpha_{t,i}(e_i + \mathbf{e} \cdot \boldsymbol{\alpha}_t)}{(\mathbf{e} \cdot \boldsymbol{\alpha}_t)(\alpha_{t,0}+1)}$
3: $s_1 \leftarrow \frac{\alpha_{t,1}(\alpha_{t,1}+1)(2e_1 + \mathbf{e} \cdot \boldsymbol{\alpha}_t)}{(\mathbf{e} \cdot \boldsymbol{\alpha}_t)(\alpha_{t,0}+1)(\alpha_{t,0}+2)}$
4: $\alpha_{t+1,0} \leftarrow \frac{m_1 - s_1}{s_1 - m_1^2}$   $\triangleright$ Compute $\boldsymbol{\alpha}_{t+1}$
5: **for** $i = 1, 2 \ldots N-1$ **do**
6:   $\alpha_{t+1,i} \leftarrow m_i \alpha_{t+1,0}$
7: $\alpha_{t+1,N} \leftarrow \alpha_{t+1,0} - \sum_{i=1}^{N-1} \alpha_{t+1,i}$
8: **return** $\boldsymbol{\alpha}_{t+1}$

---

Finally, once we have obtained $\boldsymbol{\alpha}_t$, we can compute $\mathbb{E}_{\pi_t}[w_i] = \frac{\alpha_{t,i}}{\alpha_{t,0}} = \frac{\alpha_{t,i}}{\sum_{j=1}^N \alpha_{t,j}}$. It remains only to show how to compute $\mathbb{E}[q_{s,a}|i]$ and evidence $\mathbf{e}$.

### 3.3 Algorithm

Following the *Bayesian Q-learning* framework (Dearden et al. [1998]), we model Q-values for each state-action pair as independent Gaussian distributed random variables. Since the best choice of $\Phi$ should be the optimal value function $V^*$, we model Q-values $q_{s,a}$ given the best expert $\Phi_i$ as

$$
q_{s,a}|i \sim \mathcal{N}\left(\Phi_i(s), (\sigma_{s,a}^i)^2\right),
\tag{19}
$$

where $i \in \{1, 2 \ldots N\}$. Since $(\sigma_{s,a}^i)^2$ is not known, we need to maintain an estimator of $(\sigma_{s,a}^i)^2$. However, maintaining an estimate for each expert and state-action pair would not be practical for large spaces, so we follow Downey and Sanner [2010] and replace $(\sigma_{s,a}^i)^2$ by the sample variance $\hat{\sigma}^2$ of $\mathcal{D}$. This permits constant-time updates per sample without any additional memory overhead, and this worked very well in our experiments.

Using these observations and the approximation $\pi_t = \text{Dir}(\boldsymbol{\alpha}_t)$ from Section 3.2, (9) reduces to

$$\rho_t(s, a) = \sum_{i=1}^{N} \mathbb{E}\left[q_{s,a}|i\right] \mathbb{E}_{\pi_t}\left[w_i\right] = \frac{\sum_{i=1}^{N} \Phi_i(s)\alpha_{t,i}}{\sum_{i=1}^{N} \alpha_{t,i}}, \tag{20}$$

and defines the reward shaping potential function $\hat{\Phi}$ used during training. Finally, given a return observation $d \in \mathcal{D}$ in state $s$, the evidence $e_i$ for each $i \in \{1, 2 \dots N\}$ is computed simply from the Gaussian probability distribution $\mathcal{N}\left(\Phi_i(s), \hat{\sigma}^2\right)$ in (19).

We note that all steps can be performed efficiently on-line and so this approach does not require storing $\mathcal{D}$ explicitly. Furthermore, it can be easily incorporated into general reinforcement learning algorithms without increasing the runtime complexity. Perhaps most importantly, since $\rho_t$ in (20) is a potential-based reward shaping function, it would not change the asymptotically optimal policy. The complete algorithm is summarized in Algorithm 2. Here, `TrainRL`($F$) is a general procedure for training on one state-action-reward sequence using the immediate reward function $R + F$.

---

**Algorithm 2** RL with Bayesian Reward Shaping

---

1: initialize $\boldsymbol{\alpha} \in \mathbb{R}_+^N$
2: **for** $episode = 0, 1 \dots M$ **do**                                                  ▷ Main loop
3:      $\hat{\Phi} \leftarrow \frac{\sum_{i=1}^{N} \Phi_i \alpha_i}{\sum_{i=1}^{N} \alpha_i}$                          ▷ Pool experts and compute shaped reward
4:      $F(s, a, s') \leftarrow \gamma \hat{\Phi}(s') - \hat{\Phi}(s)$
5:      $(R_t, s_t)_{t=1\dots T} \leftarrow$ `TrainRL`($F$)                      ▷ Perform one episode of training
6:      **for all** $(R_t, s_t)$ **do**                                              ▷ Posterior update
7:           update $\hat{\sigma}^2$ and compute $\mathbf{e}$
8:           $\boldsymbol{\alpha} \leftarrow$ `PosteriorUpdate`($\boldsymbol{\alpha}, \mathbf{e}$)

---

*Remarks:* Steps 3 and 4 in Algorithm 2 update the advice off-line on a sequence of cached observations. It is possible to make this algorithm on-line by performing steps 3 and 4 after each observation, but care must be taken to ensure consistency of the optimal policies (Devlin and Kudenko [2012]).

## 4  Experimental Results

In order to validate the effectiveness of our proposed algorithm, we apply it to a Gridworld problem with subgoals and the classical CartPole problem. We implement the exact tabular Q-learning and SARSA algorithms (2) and the off-policy Deep Q-Learning algorithm with experience replay (Mnih et al. [2013]). In all cases, we followed $\varepsilon$-greedy policies introduced in Section 2.2, and manually selected parameters that worked well for all experts. Policies are learned from scratch, with table entries initialized to zero and neural networks initialized randomly.

### 4.1  Gridworld

This is the 5-by-5 navigation problem with subgoals introduced in Ng et al. [1999]. We charge $+1$ points for every move, and one additional point whenever it is invalid (e.g. choosing "UP" when adjacent to the top edge, or an attempt is made to collect a flag in an incorrect order) to encourage the agent to choose valid moves. For all algorithms, we set the length of each episode to $T = 200$ steps, $\gamma = 1.0$, and $\varepsilon_t = 0.98^t$, where $t \geq 0$ is the episode.

In the tabular case, we set $\alpha = 0.4$ for Q-learning and $\alpha = 0.36$ for SARSA. The DQN is a dense network with encoded state $s$ as inputs and action-values $\{Q(s, a) : a \in \mathcal{A}\}$ as outputs, and two fully-connected hidden layers with 25 neurons per layer. We use one-hot encoding for states (see, e.g. Lantz [2013]). Hidden neurons use leaky ReLU activations and outputs use linear to allow unbounded values. The learning rate is fixed at $0.001$ throughout training that is done on-line using the Adam optimizer in batches of size 16 sampled randomly from memory of size 10000 (we found that doing 5 epochs of training per batch led to more stable convergence).

We consider the following five experts in our analysis: $\Phi_{opt}(s) = V^*(s)$ the optimal value function, $\Phi_{good}(x, y, c) = -22(5 - c - 0.5)/5$ is reasoned in Ng et al. [1999] assuming equidistant subgoals, $\Phi_{zero}(x, y, c) = 0$, $\Phi_{rand}(x, y, c) = U$ where $U \sim U[-20, 20]$, and $\Phi_{neg}(s) = -V^*(s)$.

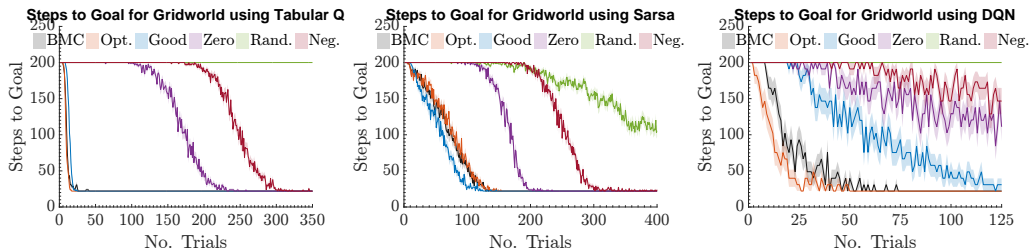

Figure 1: Test performance (number of steps required to reach the final goal) of the learned policy on Gridworld for each potential versus the number of training episodes, averaged over 100 independent runs of tabular Q/SARSA and 20 runs of DQN. BMC corresponds to Algorithm 2 applied to all potential functions.

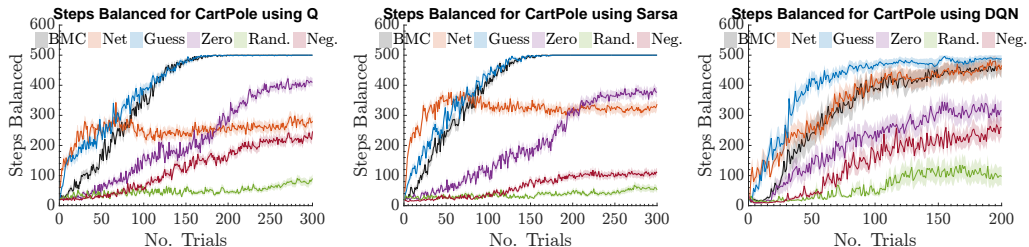

Figure 2: Test performance (number of steps that the pole was balanced) of the learned policy on CartPole for each potential versus the number of training episodes, averaged over 100 runs of tabular Q/SARSA and 20 of DQN.

## 4.2 CartPole

This is a classical control problem described in Geva and Sitte [1993] and implemented in `OpenAI Gym` (Brockman et al. [2016]). In order to encourage the agent to balance the pole, we provide a reward of $+1$ at every step as long as the pole is upright. We set $T = 500$ frames, $\gamma = 0.95$, and $\varepsilon_t = 0.98^t$. Finally, to prevent over-fitting, we stop training whenever the score attained on each of the last 5 episodes is $500$.

In both tabular cases, we set $\alpha_t = \max\{0.01, \frac{1}{2}0.99^t\}$ and $\varepsilon_t = \max\{0.01, 0.98^t\}$, where $t \geq 0$ is the episode. States $(x, \theta, \dot{x}, \dot{\theta})$ are discretized into 3, 3, 6 and 3 bins, respectively, for a total of 162 states. The neural network takes continuous inputs in $\mathbb{R}^4$ and has two fully-connected hidden layers with 12 neurons in each. Once again we use ReLU activations for hidden neurons and linear for output neurons. We set the learning rate to 0.0005 and train using the Adam optimizer. To further prevent over-fitting, we train off-line at the end of each episode on 100 batches of size 32 and use L2 regularization with penalty 1E-6.

We consider the following five experts: $\Phi_{guess}(s) = 20(1 - \frac{|\theta|}{0.2618})$ assigns a reward based on the proximity of the pole angle to the vertical, $\Phi_{net}$ is a pre-trained neural network with two hidden layers with 6 neurons per layer, $\Phi_{zero}(s) = 0$, $\Phi_{rand}(s) = U$ where $U \sim U[-20, 20]$, and $\Phi_{neg}(s) = -\Phi_{net}(s)$.

## 4.3 Summary

The performance obtained from each expert and the model combination approach are illustrated in Figure 1 for Gridworld and 2 for CartPole, and the learned expert weights are illustrated in Figure 3.

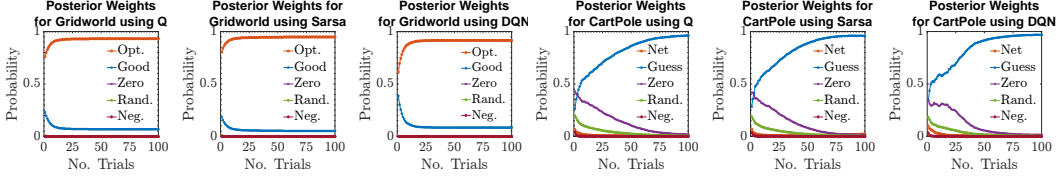

Figure 3: Posterior weights assigned to each potential as a function of the number of episodes of training, averaged over 100 independent runs using tabular Q and SARSA and 20 runs using DQN.

In Gridworld, it is interesting to see that $\Phi_{good}$ and $\Phi_{opt}$ are quantitatively very similar, yet both have similar effects on the rate of convergence in the tabular case, whereas $\Phi_{opt}$ considerably outperforms $\Phi_{good}$ in the deep learning case. As shown in Figure 3, our algorithm assigns most of its weight to $\Phi_{opt}$ and results in near-optimal performance in all three cases.

In CartPole, it is not immediately clear that $\Phi_{guess}$ is better than $\Phi_{net}$, since both should be very close to $V^*$. However, $\Phi_{net}$ is both a biased estimate of $V^*$ and noisy (due the inexactness of gradient descent), whereas the simple expert $\Phi_{guess}$ is highly related to the goal (keeping the pole centered). Furthermore, $\Phi_{net}$ is even less accurate in the tabular case due to state discretization. Once again, Figure 3 clearly shows that our approach can handle both analytic and computational advice and is sensitive to approximation error and noise.

## 5    Conclusion and Future Work

In this paper, the decision maker is presented with multiple sources of expert advice in the form of potential-based reward functions, some of which can be misleading and should not be trusted. We assumed that the decision makes does not know a priori which expert(s) to trust, but rather learns this from experience in a Bayesian framework. More specifically, we followed the Bayesian model combination approach and assigned posterior probabilities to distributions over experts. We showed that the total expected return is a linear combination of individual expert predictions, weighted by the posterior beliefs assigned to them. We solved the issue of tractability by projecting the true posterior distribution onto the Dirichlet family using moment matching, and then specialized our analysis to Bayesian Q-learning. Our approach followed the potential-based reward shaping framework and does not change the optimal policies. Finally we showed that our proposed method accelerated the learning phase when solving discrete and continuous domains using different learning algorithms.

Further extensions and generalizations of this work could include rigorous theoretical analysis of posterior convergence under certain conditions on the reward shaping functions. It is also possible to extend our analysis to state/action-dependent weightings of experts, at the cost of higher space complexity; this could be useful in situations where the most suitable potential function changes in different regions of the state space. It also remains to scale our work to large-scale and real-world problems where imperfect advice and issues in convergence could be more prevalent.

### Acknowledgments

We would like to thank the NeurIPS reviewers for their feedback, that significantly improved this paper.

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
