[Reviews · NeurIPS 2018]

Reviewer 1



The paper describes a new algorithm to leverage domain knowledge from several experts in the form of reward shaping. These different reward shaping potentials are combined through a Bayesian learning technique. This is very interesting work. Since domain knowledge might improve or worsen the convergence rate, the online Bayesian learning technique provides an effective way of quickly identifying the best domain knowledge by gradually shifting the posterior belief towards the most accurate domain knowledge. At a high level, the approach makes sense. I was able to follow the derivations. I have a few comments. 1) The section on moment matching is good, however the moments that are matched (first and second order moments) do not follow from minimizing KL-divergence. Minimizing KL-divergence corresponds to matching the "natural" moments, which in the case of the Dirichlet are E[log theta_i] for all i. See Section 4.2.1 of Farheen Omar's PhD thesis for a derivation and explanation (https://cs.uwaterloo.ca/~ppoupart/students/farheen-omar-thesis.pdf). 2) The symbol R is used to denote the immediate reward in Sections 2.1 and 2.3, while it is also used to denote the expected Q-values in Sections 2.2 and 3. This is confusing. I recommend to use different symbols. 3) Is the proposed Bayesian reward shaping technique guaranteed to preserve the optimal value function? The paper highlights that reward shaping should satisfy Equations 4 and 5. Does the proposed Bayesian reward shaping scheme satisfy those equations? If yes, can you include a theorem? If not, can you include a discussion to that effect? 4) The experiments are minimal, but perhaps sufficient. They demonstrate the effectiveness of the proposed reward shaping on a toy grid world (discrete states) and the cart-pole problem (continuous states). Could you show additional experiments in domains with sparse rewards where the proposed reward shaping technique could lead to improved state-of-the-art results? This could make a big difference between just publishing the idea vs showing the world that this is a useful tool to improve the state of the art in some challenging domains.

Reviewer 2



Summary: Reward shaping can speed up learning, and this paper considers the setting where experts propose functions of unknown quality. The algorithm combines the information of the experts to find an optimal ensemble without computational cost with respect to standard TD methods. This idea can be incorporated on any training procedure based on state-action-reward sequences. Simulations show the effectiveness of the approach. Specific comments: 1) It would be useful to number individual lines in equation (7) that are refered to in the lines below. 2) There is still quite a bit of noise in the outcomes of simulations (Fig 1 and 2). If feasible it might be helpful to average over more trials. 3) It is unclear whether code to replicate the simulations will be made available. Making high quality code available for others would be a big plus. 4) A discussion on where experts can come from would be a useful addition, as it might not always be obvious how to do this. In particular, I am wondering if this algorithm can be useful in the following scenario: consider a complicated RL task, which requires a rich model to be able to succeed. However, training this model from scratch would take a long time due to its complexity. Instead, smaller models are first trained to perform smaller tasks. By combining these as experts, the richer model is able to learn much faster in its initial stages. 5) The peaper and simulations seem to focus on experts that are either uniformly good or uniformly bad. One can imagine two experts that are useful in different parts of the state space. The authors might want to comment on this scenario. Overall: The paper is well written, easy to follow, and provides a clean algorithm. It is self-contained, and the simulations are convincing. If high quality code is made available alongside the submission that would make a big difference (please comment). -- Update after author response: Thanks for the detailed response, which only reinforces my opinion that the paper should be accepted. I agree with other reviewers that the simulations are meager and look forward to seeing the additional simulations in the final version. Good luck!

Reviewer 3



UPDATE: I have read the authors' response, and it was particularly helpful to me in showing that I misinterpreted some of the reward modeling. I now understand better that the learning from experts is actually learning potentials, and yes in that case there is a theoretical convergence guarantee, so those parts of my criticism have been addressed. I do feel there is still an experimental gap here though, because no other approaches from the Bayesian RL or other literature were tested against,a and the number of variants of the current algorithm tested are relatively small and in toy domains. Summary: The paper describes a Bayesian method for combining the potential functions of multiple experts over time in a reinforcement learning scenario. A method is given for updating both the posterior probabilities of the potentials and calculating the weighted reward function used in choosing an action. Empirical results in a gridworld and cart pole are given. Review: I think the problem setting the authors have chosen is very rich and poses an interesting problem for a reinforcement learning agent. I also like that the expert advice is provided in the form of potentials, rather than trajectories or exact reward functions, because it makes the objective clearly defined (maximize the agent’s own reward) and the role of the experts (to speed up exploration) clear as well. However, I thought the solution wasn’t quite worked out or tested to the level of detail needed for a NIPS paper and comparisons to other Bayesian approaches were not attempted. Specifically, the paper could be substantially improved based on the following: The algorithm itself is Bayesian in that given the data already collected, it calculates a posterior on the potential expert potentials. But the algorithm does not really make use of that knowledge in its action selection. That is, the weighted potentials do certainly affect exploration, but the algorithm does not take actions to explore *which* expert might be right, which is a hallmark of Bayesian RL algorithms. A good, and very relevant, example of such an algorithm is the BOSS algorithm from “A Bayesian Sampling Approach to Exploration in Reinforcement Learning” (UAI 2009). That algorithm has a space of potential models, and performs exploration in such a way that either explicitly refutes some models or follows an optimal strategy. One could perform similar exploration of the potentials in order to better explore the weight space, and I was surprised to not see a Bayesian RL approach here that did so. It seems like a comparison to a Bayesian RL algorithm that treats the experts as potential models is needed. In addition, the empirical results did not cover enough scenarios to be completely convincing. The current results definitely show the mixed potentials are quickly converging to the performance of a good potential function. But what if the mixture did not contain a good potential? Or contained 9 bad functions and 1 good one? How does the performance of the algorithm vary as that good/bad mixture varies? If this algorithm is going to be used in larger environments, there will likely be many more experts and potentials, and we need to know how the algorithm will behave under these conditions. From a theoretical perspective, the paper lacks a rigorous theoretical result proving convergence or ensuring optimal asymptotic behavior. Without such guarantees, and only the limited experiments, the algorithm does not feel like it has been proven enough. I also found the result at the end of Section 3.1 confusing. Doesn’t the result that the true rewards can be found by a weighted combination of the experts require that the value function be in the convex hull of their individual functions? Why is that assumption not stated outright? Again, it is unclear how the algorithm will behave with all very incorrect potentials. Minor points: The paper makes a number of very general claims about how it can be applied with any exploration policy, any RL algorithm, etc., but then goes on to state there are no theoretical guarantees and provides an implementation algorithm that may only be applicable for model free learning. Would Rmax work like this? I think the general claims may be a little overreaching, though I think a large class of algorithms are indeed covered. Lines 142-145 make reference to equation numbers that do not appear above The derivation in lines 141-142 is somewhat hard to follow and could be improved by including a numeric example of the computation.